# Engagement of the G3BP2-TRIM25 Interaction by Nucleocapsid Protein Suppresses the Type I Interferon Response in SARS-CoV-2-Infected Cells

**DOI:** 10.3390/vaccines10122042

**Published:** 2022-11-29

**Authors:** Zening Yang, Jing Li, Jiali Li, Huiwen Zheng, Heng Li, Qingrun Lai, Yanli Chen, Li Qin, Yuanyuan Zuo, Lei Guo, Haijing Shi, Longding Liu

**Affiliations:** 1Chinese Academy of Medical Sciences, Key Laboratory of Virus Vaccine Research & Development System Innovation, Institute of Medical Biology, Chinese Academy of Medical Sciences & Peking Union Medical College, Kunming 650031, China; 2Key Laboratory of Systemic Innovative Research on Virus Vaccine, Chinese Academy of Medical Sciences, Kunming 650031, China; 3Kunming Children’s Hospital, Kunming 650225, China

**Keywords:** SARS-CoV-2, nucleocapsid protein, G3BP2, TRIM25, RIG-I-like pathway

## Abstract

The nucleocapsid (N) protein contributes to key steps of the SARS-CoV-2 life cycle, including packaging of the virus genome and modulating interactions with cytoplasmic components. Expanding knowledge of the N protein acting on cellular proteins and interfering with innate immunity is critical for studying the host antiviral strategy. In the study on SARS-CoV-2 infecting human bronchial epithelial cell line s1(16HBE), we identified that the N protein can promote the interaction between GTPase-activating protein SH3 domain–binding protein 2 (G3BP2) and tripartite motif containing 25 (TRIM25), which is involved in formation of the TRIM25-G3BP2-N protein interactome. Our findings suggest that the N protein is enrolled in the inhibition of type I interferon production in the process of infection. Meanwhile, upgraded binding of G3BP2 and TRIM25 interferes with the RIG-I-like receptor signaling pathway, which may contribute to SARS-CoV-2 escaping from cellular innate immune surveillance. The N protein plays a critical role in SARS-CoV-2 replication. Our study suggests that the N protein and its interacting cellular components has potential for use in antiviral therapy, and adding N protein into the vaccine as an antigen may be a good strategy to improve the effectiveness and safety of the vaccine. Its interference with innate immunity should be strongly considered as a target for SARS-CoV-2 infection control and vaccine design.

## 1. Introduction

SARS-CoV-2 propagates via infectious particles with a 30-kb single-stranded RNA genome, which encodes 16 nonstructural proteins, five major open reading frames for nonstructural replicase polyproteins and structural proteins, namely, spike (S), envelope (E), membrane (M), and nucleocapsid (N) [1,2]. Among these viral proteins, the N protein contributes to multiple steps of the viral life cycle, including packaging of the RNA genome, regulating viral RNA synthesis and modulating interactions with cellular components in the cytoplasm [3]. High levels of IgG antibodies against N have been detected in both SARS-CoV and SARS-CoV-2 patients [4,5]. A study shows that memory T cell responses to N proteins are persistent in recovered patients with SARS-CoV and SARS-CoV-2 [6]. Thus, the N protein is considered as a representative antigen for specific T cell proliferation and cytotoxic activity [7]. Some recent studies have found that some specific antibodies against the N protein also have a neutralizing activity, so some small molecular polypeptide drugs targeting specific epitopes of N protein are also being developed; similarly, these clues are also very helpful to the development of vaccines [8]. In addition to the adaptive immune response to SARS-CoV-2 infection, innate immunity triggered by the RIG-I-like receptor signaling pathway is important in defense against RNA viruses such as coronaviruses [9,10]. Previous studies on SARS-CoV suggest that viral components including both the M protein and N protein impair type I interferon production by suppressing the RIG-I-like receptor pathway through different mechanisms [11]. Recently, several studies have revealed that the N protein interacts with the tripartite motif containing 25 (TRIM25) to inhibit activation of retinoic acid-inducible gene-I (RIG-I) [12,13].

TRIM25 plays an important role in the activation of the RIG-I receptor pathway via binding with the caspase recruitment domain (CARD) of RIG-I and resulting in ubiquitin activation signal transmission of RIG-I [14,15]. Within the signal transduction regulator family, GTPase-activating protein SH3 domain–binding protein 1 (G3BP1) and protein 2 (G3BP2) are closely related to innate immunity via the formation of stress particles [16,17,18,19,20]. G3BP2 is involved in the nuclear translocation of NF-κB [14,15] and has been identified as a novel interacting partner with TRIM25 in prostate cancer cell proliferation [21]. Recent data show that SARS-CoV-2 inhibits stress particle formation by interacting with G3BPs during replication to promote the separation of N from viral RNA [18].

Many studies have shown that G3BP1 and G3BP2 stress particles are crucial for the host antiviral process [22]. In this study, we found G3BP2 to be consistently involved in this process via the formation of the TRIM25-G3BP2-N protein interactome. G3BP2 is also known as a stress granule assembly factor in the cytoplasm for viral replication complexes during infection of some RNA viruses [23]. Other studies have revealed a close relationship between G3BP2 and the NF-κB signaling pathway of innate immunity [20]. We speculate that the N protein promotes G3BP2 and TRIM25 binding in the process of SARS-CoV-2 infection, interfering with RIG-I-like receptor signaling, weakening the innate immune response, and leading to SARS-CoV-2 immune escape. Our study further explains the mechanism by which N inhibits interferon I production, suggesting that It has potential as a vaccine antigen and an antiviral therapeutic target.

## 2. Materials and Methods

### 2.1. Cells and Viruses

16HBE, HEK293T and A549 cells were obtained from the American Type Culture Collection (ATCC, Manassas, VA, USA) and cultured in Dulbecco’s Modified Eagle’s medium (DMEM) (Biological Industries) supplemented with 10% fetal bovine serum (FBS) (ThermoFisher, Waltham, MA, USA) and 1% penicillin–streptomycin. The viral strain SARS-CoV-2-KMS1/2020 (GenBank accession number: MT226610.1) was isolated from sputum collected from a COVID-19 patient by IMBCAMS, and propagated and tittered on Vero cells in DMEM supplemented with 2% FBS. The harvested virus was purified by Sepharose 6 Fast Flow chromatography. The virus stock titer was 10^6^ TCID50/mL and the concentration was diluted to a suitable working concentration before the experiment. All work with infectious SARS-CoV-2 was approved under biosafety level-4 (BSL-4) conditions by the Institutional Biosafety Committee of IMB.

### 2.2. CCID50 Assay

The culture of Vero cells with 96-well plates and the SARS-CoV-2 virus was diluted by 10 times with MEM medium containing 2% serum. The cells were infected with each dilution. We observed the CPE phenomenon 7 days after virus infection, and the virus titer was calculated according to the Reed & Muench method [24].

### 2.3. Plasmids and Reagents

Gene fragments of the SARS-CoV-2 protein were obtained by PCR of SARS-CoV-2 cDNA and cloned into the pAcEGFP-C1 vector (TAKARA, Tokyo, Japan). G3BP2 was cloned into the pcDNA3.1 vector (ThermoFisher, Waltham, MA, USA). TRIM25 was cloned into the pCMV-Myc vector (TAKARA, Tokyo, Japan). All DNA transfections were performed using FuGENE HD (Promega, Madison, WI, USA). Poly (I:C) transfection was performed using Lipofectamine 2000 (ThermoFisher, Waltham, MA, USA) and siRNA transfection using Lipofectamine RNAimax (ThermoFisher, Waltham, MA, USA).

### 2.4. Antibodies

Anti-NF-κB, anti-IRF3, anti-TBK1, anti-IFN-β, anti-GAPDH, anti-pNF-κB, anti-pTBk1, anti-pIRF3 and anti-DDDDK for WB and immunofluorescence (IF) assays were purchased from Abcam (Abcam, Cambridge, UK). Anti-HA and anti-Myc for WB and immunofluorescence (IF) assays were purchased from Thermo Fisher (ThermoFisher, Waltham, MA, USA). Anti-J-2 for IF assays were purchased from Scicons (Scicons, Szirák, Hungary). Secondary antibodies labeled with Alexa Fluor 488, Alexa Fluor 555, Alexa Fluor 594 or Alexa Fluor 647 for the IF assay were purchased from Abcam (Abcam, Cambridge, UK). Goat anti-mouse IgG H&L (HRP) and goat anti-rabbit IgG H&L were purchased from Abcam (Abcam, Cambridge, UK) and used in the WB assays.

### 2.5. Immunofluorescence Analysis

The cells were cultured in a 12-well plate with cover slides according to the density of each well of 2 × 10^5^. Twenty-four hours later, the cells were infected with SARS-CoV-2 (MOI = 0.5) or transfected with 3 μg of plasmid. Cell samples were collected according to the experimental design. After washing with PBS, the cells were fixed with 4% paraformaldehyde for 15 min, blocked with 5% BSA, and permeabilized with 0.1% Triton for 45 min. The treated cells were incubated with the primary antibody at room temperature for 2 h and then with the secondary antibody was incubated for 1 h. Glass coverslips on the slides were sealed by dripping the DAPI stain and sealing agent. Cell samples were analyzed by a Leica SPR8 laser confocal microscope. For FLIM/FRET experiments, the “donor only” sample was transfected with G3BP2 plasmids. We analyzed the colocation between G3BP2 and TRIM25 with N protein exist or not. The average fluorescence lifetimes of the donor-only and FRET samples were measured in fast FLIM mode. The measurement was completely controlled via Leica Application Suite Advanced Fluorescence. The FRET efficiency was calculated from the ratio of the FRET donor lifetime *τ_quench_* and the non-FRET lifetime *τ_quench_* as follows:

*FRET efficiency* (%) = 1 − (*τ_donor_, FRET*)/(*τ_donor_, no FRET*)
(1)


### 2.6. Western Blot Analysis

16HBE, HEK293T and A549 cells were preplated into six-well plates at a density of 5 × 10^5^ cells per well. The next day, the cells were inoculated with SARS-CoV-2 (MOI = 0.5), transfected with 3 μg of plasmid (using FuGENE HS transfection reagent) or treated with poly (I:C). After collection at the appropriate time points, total cellular protein was extracted with an RIPA (Thermo Scientific, Waltham, MA, USA) buffer consisting of 25 mM Tris·HCl (pH 7.6), 150 mM NaCl, 1% NP-40, 1% sodium deoxycholate, and 0.1% SDS complete EDTA-free protease inhibitors. Following SDS–PAGE, Transfer of nitrocellulose membrane by wet rotary film meter (GenScript, Nanjing, China) under the following common conditions. The membranes were blocked at room temperature for 1 h and incubated with primary antibodies diluted with 5% skim milk overnight at 4 °C, washed with 0.1% PBST, incubated with secondary antibody diluted with 5% skim milk at room temperature for 1 h, and washed with 0.1% PBST. ECL chemiluminescence was examined using a BIO-RAD (BIO-RAD, Hercules, CA, USA) imager, and the grayscale results were analyzed using a Gel-Pro analyzer.

### 2.7. Pulldown and Immunoprecipitation Analysis

293T cells were placed in 10-cm culture dishes and transfected with pAcGFP and pAcGFP-N-Flag constructs 12 h later. The cell samples were collected after 24 h; total protein was extracted with a Pierce™ IP buffer (Thermo Scientific, Waltham, MA, USA) consisting of 25 mM Tris·HCl (pH 7.4), 150 mM NaCl, 1% NP-40, 1 mM EDTA, and 5% glycerol complete EDTA-free protease inhibitors. The protein samples were immunoprecipitated using Pierce Anti-DYKDDDDK Magnetic Agarose (Thermo Scientific, Waltham, MA, USA) at room temperature for 20 min. One hundred microliters of SDS–PAGE sample buffer was then added to the centrifuge tube containing the incubated agarose magnetic beads, and the mixture was gently vortexed and incubated at 95–100 °C for 5–10 min. Following denaturation, proteins were detected by western blotting.

### 2.8. Knockdown

siRNA targeting G3BP2 was purchased from Thermo Fisher. 16HBE and A549 cells were seeded into a 12-well plate at 2 × 10^5^ cells per well and transfected with 30 pg siRNA; cell samples were collected after 48 h. Total RNA was extracted with TRIzol (TIANGEN, Beijing, China), and a One-Step TB Green ^®^PrimeScript RT–PCR Kit (TAKARA, Tokyo, Japan) and Bio-rad CFX Detection System was used for RT–PCR.

### 2.9. RT–PCR Analysis

16HBE and A549 cells were seeded into a 12-well plate at 2 × 10^5^ cells per well and infected with SARS-CoV-2 virus (MOI = 0.5) or transfected with poly (I:C) (Sigma–Aldrich, St. Louis, MO, USA). The cell samples were collected after 0, 3, 6 and 24 h, and the total RNA of cell samples were extracted with TRIzol (TIANGEN, Beijing, China). One-Step TB Green ^®^PrimeScript RT–PCR Kit (TAKARA, Tokyo, Japan) and Bio-rad CFX Detection System were used for RT–PCR. The transcription levels of target genes were quantified by the 2^−ΔΔct^ [25]. The results were normalized to GAPDH expression.

### 2.10. Statistical Analyses

Each experiment described in this manuscript was repeated at least three times. All statistical analyses were performed with GraphPad Prism software (version 5, La Jolla, CA, USA), and a *t*-test was used for the statistical analyses. The data obtained from all experiments are presented as means ± standard deviations (SDs). Statistical significance between different groups was calculated by a two-tailed unpaired Student’s *t*-test or two-way ANOVA test. All the data used for statistical analysis were tested by the Shapiro-Wilk method, and the data were consistent with normal distribution. A *p* value of less than 0.05 was assumed to indicate statistical significance.

## 3. Results

### 3.1. The N Protein Contributes to Inhibition of the RIG-I-like Receptor Signaling Pathway during SARS-CoV-2 Replication in 16HBE Cells

Previous studies have shown that viral proteins of SARS-CoV can inhibit type I interferon in the RIG-I-like receptor signaling pathway. In our study of 16HBE cells infected with SARS-CoV-2, we found that the level of CXCL10 transcription increased gradually with virus infection progression but that relative expression of ISG15, NF-κB, IFN-α and IFN-β increased within 6 h and then decreased (Figure 1A). Moreover, when detecting downstream regulatory proteins of the RIG-I-like receptor pathway, we observed that phosphorylation of IRF3, NF-κB and TBK1 increased significantly after virus infection but decreased at 24 h.p.i. (hours post-infection) (Figure 1B). Viral RNA or double-stranded RNA can initiate RIG-I-like receptor signaling in response to infection, and the synthesis of viral negative-strand RNA has been proven in several studies to be a critical stage for SARS-CoV replication. Notably, by using J-2 staining, we observed increases in double-stranded RNA in 16HBE cells during the infection process at 3, 6 and 24 h.p.i., accompanied by viral S and N expression (Figure 1C). Thus, we speculate that SARS-CoV-2 has an inhibitory effect on the type I IFN response in the viral replication stage, even though IFN-α/IFN-β was upregulated in the early stage of infection in cells. The findings suggest that SARS-CoV-2 infection activates the innate immune response at the initial stage, but the virus gradually inhibits the RIG-I-like receptor signaling pathway.

To investigate the function of viral components in this inhibition, we constructed several expression plasmids for SARS-CoV-2 major structural proteins and open reading frames. We found that the M protein and N protein had inhibitory effects on the production of IFN-β (Figure 2A). Compared to poly (I:C), the N protein significantly reduced the relative expression of ISG15 and IFN-β in stimulated cells (Figure 2B). However, CXCL10 transcription following poly (I:C) activation was not reduced by N protein pre-transfection, which may result from the strong triggering effect of poly (I:C) on the production of inflammatory cytokines and chemokines such as TNF-α, IL-6, and CXCL10. Furthermore, N reduced the level of IRF3, TBK1 and NF-κB phosphorylation (Figure 2C). These results indicate that the N protein of SARS-CoV-2 inhibits the activation of the RIG-I-like receptor signaling pathway.

### 3.2. Cellular TRIM25 and G3BP2 Proteins Are Coimmunoprecipitated by the N Protein at Its C-Terminus

The N proteins of SARS-CoV and MERS were reported to inhibit activation of the RIG-I-like receptor signaling pathway via interacting with TRIM25. In our study, based on mass spectrometry (MS) analysis on interacting cellular proteins from coimmunoprecipitation experiments. (Appendix A), we found the TRIM25 peptide as well as a peptide of protein G3BP2 co-interacting with the N protein. We verified the MS results in Appendix A by western blotting (Figure 3A) and detected a direct interaction between the N protein and TRIM25 and G3BP2 (Figure 3B,C), revealing an interaction between TRIM25 and G3BP2 (Figure 3D). Immunofluorescence also showed an interaction between the N protein and G3BP2 (Figure 3E). Moreover, we constructed expression plasmids of N deletion (amino acids 1–174, 175–255 and 255–419) to explore regions interacting with TRIM25 and G3BP2 (Figure 3F). We found that TRIM25 interacted with N protein amino acids 1–174 and 255–419 (Figure 3G) and that G3BP2 interacted with amino acids 255–419 (Figure 3H). The above results show that the C-terminus of the SARS-CoV-2 N protein interacts directly with both TRIM25 and G3BP2.

### 3.3. Engagement of G3BP2-TRIM25 Binding Is Enhanced by N Protein Recruitment

The above findings suggest that the N protein may recruit TRIM25 and G3BP2 to enhance the interaction between them. We observed that TRIM25 binding to G3BP2 increased significantly in 293T cells pre-transfected by Flag-N (Figure 4A). Furthermore, this interacting was observed more significantly in the 293T cells co-transfected with Myc-TRIM25, HA-G3BP2 and Flag-N (Figure 4B). Confocal imaging also revealed that TRIM25 and G3BP2 gradually colocalized with N in the cytoplasm of SARS-CoV-2-infected cells from 0 to 24 h.p.i. (Figure 4C). For further verifying G3BP2-TRIM25 interaction enhanced by the N protein, we designed a fluorescence resonance energy transfer (FRET)/fluorescence lifetime imaging (FLIM) assay to interpret this process. In 293T cells co-transfected with Myc-TRIM25, HA-G3BP2 and Flag-N plasmids, we observed three protein co-localization regions and aggregation as shown in the red circle (Figure 4D). The proximity between two proteins in cells can be measured by calculating the FRET efficiency values. We found that the fluorescence energy transfer from G3BP2 to TRIM25 was significantly enhanced by the N protein being co-expressed in cells, and the FRET efficiency values were 42%, compared with 36% in cells without N protein expression. Our results confirmed G3BP2 binding TRIM25 in cytoplasm, and N protein can enhance this interaction (Figure 4E). Interestingly, we also found that N promoted the expression of both G3BP2 and TRIM25 (Appendix A).

### 3.4. Colocalization of G3BP2, TRIM25 and N Protein in Respiratory Tract Tissue Was Observed in SARS-CoV-2 Infection of Rhesus Monkeys

We obtained similar findings in vivo in a nasal infection experiment of rhesus monkeys. Using tissue sections of the nasopharynx and lungs of rhesus monkeys infected with SARS-CoV-2 for 3 days, we detected TRIM25, G3BP2 and N protein colocalization to form a complex (Figure 5A). There were numerous collocated regions of the three proteins in the lungs (Figure 5B). Interestingly, on the seventh day after infection, the TRIM25 protein in nasopharyngeal tissue decreased significantly, with only G3BP2 and N colocalizing (Figure 5C). Compared with the cellular infection experiment, the protein expression of G3BP2 and TRIM25 increased gradually in the early stage of virus infection. However, in the later stage of infection, the expression of TRIM25 decreased to below the normal level, which was different from that of G3BP2.

### 3.5. Overexpression of G3BP2 Impairs TRIM25 Regulation of the Type I Interferon Signaling Pathway during SARS-CoV-2 Infection

The above results suggest that G3BP2 is involved in the process of SARS-CoV-2 N protein binding to TRIM25 and may play a critical role in the antiviral response. In A549 cells pre-transfected with the HA-G3BP2 plasmid, gene expression of ISG15 and IFN-β was inhibited after poly (I:C) stimulation (Figure 6A). At the protein level, overexpression of G3BP2 affected phosphorylation of IRF3 and NF-κB (Figure 6B) and bound with intracellular TRIM25 (Figure 6C). In addition, the SARS-CoV-2 virus titer in 16HBE cells overexpressing G3BP2 was higher than that of control cells at 48 h.p.i. (Figure 6D). The above results implied that G3BP2 functioning in inhibition of the RIG-I-like receptor pathway may be related to binding to TRIM25. To verify this hypothesis, we knocked down G3BP2 by siRNA and found that ISG15 and CXCL10 increased significantly in cells, but expression of IFN-β was still inhibited (Figure 6E). We also observed a significant increase in phosphorylation levels of IRF3, TBK1 and NF-κB in cells with G3BP2 knockdown (Figure 6F). Similarly, SARS-CoV-2 replication was increased in G3BP2-knockdown 16HBE cells (Figure 6G), and expression of the IFN-β gene was inhibited. Finally, we detected the activation of the RIG-I signaling pathway with the simultaneous overexpression of RIG-I, TRIM25 and G3BP2, and found that the presence of G3BP2 still inhibits the activation of the RIG-I pathway (Figure 6H). Based on the above results, we believe that G3BP2 plays a role in type I interferon production, and its binding to TRIM25 inhibits activation of the RIG-I-like receptor pathway.

## 4. Discussions

Compared with SARS-CoV and MERS-CoV, SARS-CoV-2 is highly transmissible and causes serious respiratory symptoms. As many people infected with SARS-CoV-2 do not have COVID-19 symptoms, there are to date insufficient data to explain this phenomenon. Our previous study of nasal infection in rhesus monkeys revealed a pattern of long-term viral shedding related to disturbed innate immune responses in the respiratory tract [26]. Additionally, in previous studies, we found that SARS-CoV-2 has the ability to duplicate and is maintained in 16HBE cells for longer than 7 days, with no obvious cytopathic effect (CPE) and accompanied by a modified transcription profile of innate immune signaling molecules [27]. Different from other RNA viruses, the interferon response induced by SARS-CoV-2 is notably weak [28,29,30], which may be related to its multiple inhibition of the cellular innate immune response, including several viral proteins that affect the production of type I interferon during replication in cells [31,32,33,34].

Coronavirus structural proteins, the M protein and N protein, were reported to impair type I interferon production. Previous studies have shown that the N protein of SARS-CoV interacts with TRIM25 to inhibit activation of the RIG-I-like receptor signaling pathway [12]. In our study, we verified that the SARS-CoV-2 N protein interacts with TRIM25 and inhibits the RIG-I-like receptor pathway in 16HBE cells. In contrast to previously published results showing that the C-terminus of SARS-CoV N interacts with TRIM25, we detected two regions of the N protein that bind to TRIM25. Hence, the SARS-CoV-2 N protein has a stronger binding affinity to TRIM25, which may result in the enhanced inhibition of RIG-I-like receptor pathway activation.

One of the most important findings of this study is that G3BP2 is involved in the process of N protein binding to TRIM25. The presence of the N protein promotes TRIM25 and G3BP2 binding. We further demonstrate that the N protein has a certain recruitment effect on TRIM25 and G3BP2 through a shared binding region at the C- and N-termini. Clued by reports that showed that G3BP2 protein knockout inhibits entry of NF-κB into the nucleus [20], we found that the production of type I interferon is affected by G3BP2 knockdown in cells, demonstrating that G3BP2 is an important factor in innate immunity during SARS-CoV-2 infection. The interaction between G3BP2 and TRIM25 driven by the N protein leads to negative feedback regulation, which may be the reason why the RIG-I-like receptor pathway is inhibited when G3BP2 is overexpressed and interacts with TRIM25.

## 5. Conclusions

In general, on the basis of previous studies, our results further explain how the N protein of SARS-CoV-2 inhibits the activation of the RIG-I-like receptor signal pathway. The results further clarify the importance of the N protein in the process of SARS-CoV-2 infection, and suggest that the N protein has great potential in the development of vaccines and could be a good strategy for increasing the effectiveness of SARS-CoV-2 vaccines in the future. We drew a hypothetical mechanism model diagram to demonstrate our results (Figure 7).

## Figures and Tables

**Figure 1 vaccines-10-02042-f001:**
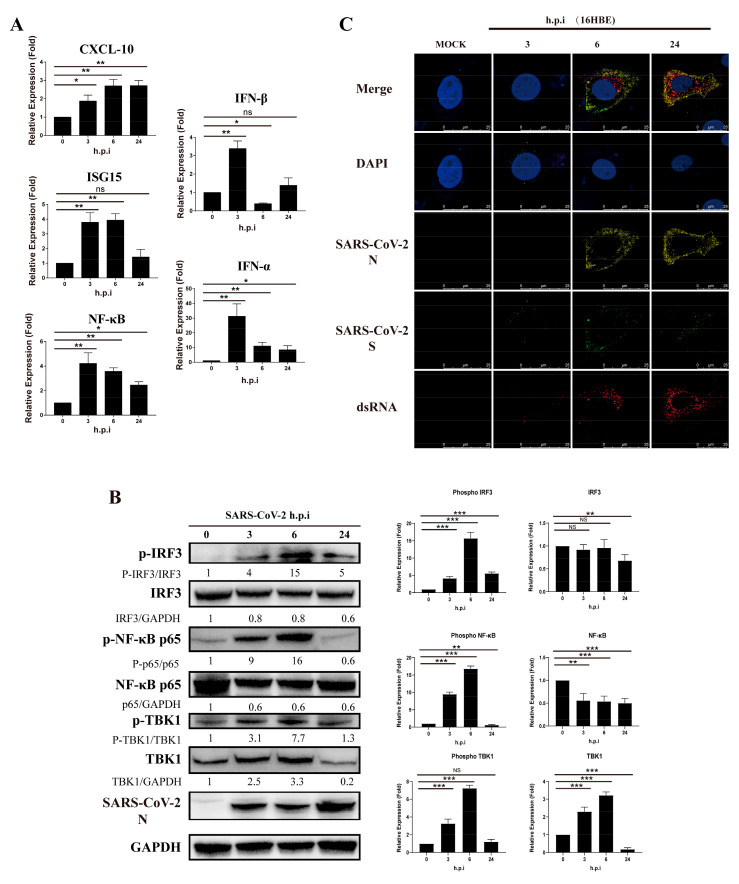
SARS-CoV-2 inhibits activation of IFN-α/β and related proteins in 16HBE cells during viral replication. (**A**) Transcription level of RIG-I-like receptor signaling pathway genes in 16-HBE cells at 0, 3, 6, and 14 h post-SARS-CoV-2 infection. Data represent the relative expression of CXCL10, ISG15, NF-κB, IFN-α and IFN-β are presented as means ± SD (paired *t* test, *n* = 3, biological replicates per group, ns > 0.05, * 0.01 < *p* < 0.05, ** 0.001 < *p* < 0.01, *** *p* < 0.001). (**B**) Detection of phosphorylation by western blot analysis of the IRF3, NF-κB and TBK1 proteins. Grayscale results were analyzed using a Gel-Pro analyzer. Data are presented as means ± SD (paired *t* test, *n* = 3 biological replicates per group, ns > 0.05, * 0.01 < *p* < 0.05, ** 0.001 < *p* < 0.01, *** *p* < 0.001). (**C**) Double-strand RNA stained by J-2 in 16HBE cells during the SARS-CoV-2 infection process at 3, 6 and 24 h.p.i. Red fluorescence corresponds to a double-stranded RNA-positive signal, yellow and green fluorescence labels SARS-CoV-2 N and S expression, respectively, and blue fluorescence corresponds to nuclei (DAPI staining; scale bar, 25 μm).

**Figure 2 vaccines-10-02042-f002:**
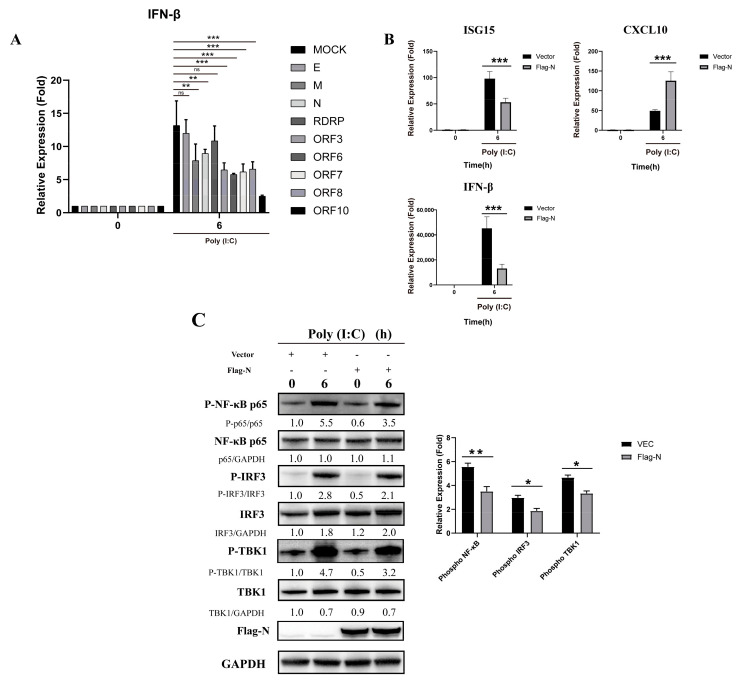
The N protein interferes with the RIG-I-like receptor signaling pathway in HEK293T cells. (**A**) The pcDNA3.1 vector and pcDNA3.1-E, M, N, RDRP, ORF3, ORF6, ORF-7, ORF-8, and ORF-10 were transfected into cells. Twenty-four hours after transfection, the transfected cells were stimulated with poly (I:C). Total RNA was extracted from 0- and 6-h cell samples for real-time PCR. Data represent the relative expression of IFN--β. (paired t test, n = 3, biological replicates per group, ns > 0.05, * 0.01 < *p* < 0.05, ** 0.001 < *p* < 0.01, *** *p* < 0.001). (**B**) The pcDNA3.1 vector and pcDNA3.1-N were transfected into cells. Twenty-four hours after transfection, the transfected cells were stimulated with poly (I:C). Total RNA was extracted from 0- and 6-h cell samples for real-time PCR. Data represent relative expression of IFN-β. Data are presented as means ± SD (paired *t* test, *n* = 3 biological replicates per group, ns > 0.05, * 0.01 < *p* < 0.05, ** 0.001 < *p* < 0.01, *** *p* < 0.001). (**C**) The pcDNA3.1 vector and pcDNA3.1-N-Flag were transfected into cells. Twenty-four hours after transfection, the transfected cells were stimulated with poly (I:C). Phosphorylation was detected by western blot analysis of IRF3, NF-κB and TBK1 proteins from 0- and 6-h cell samples. Grayscale results were analyzed using a Gel-Pro analyzer. Data are presented as means ± SD (paired *t* test, *n* = 3 biological replicates per group, ns > 0.05, * 0.01 < *p* < 0.05, ** 0.001 < *p* < 0.01, *** *p* < 0.001).

**Figure 3 vaccines-10-02042-f003:**
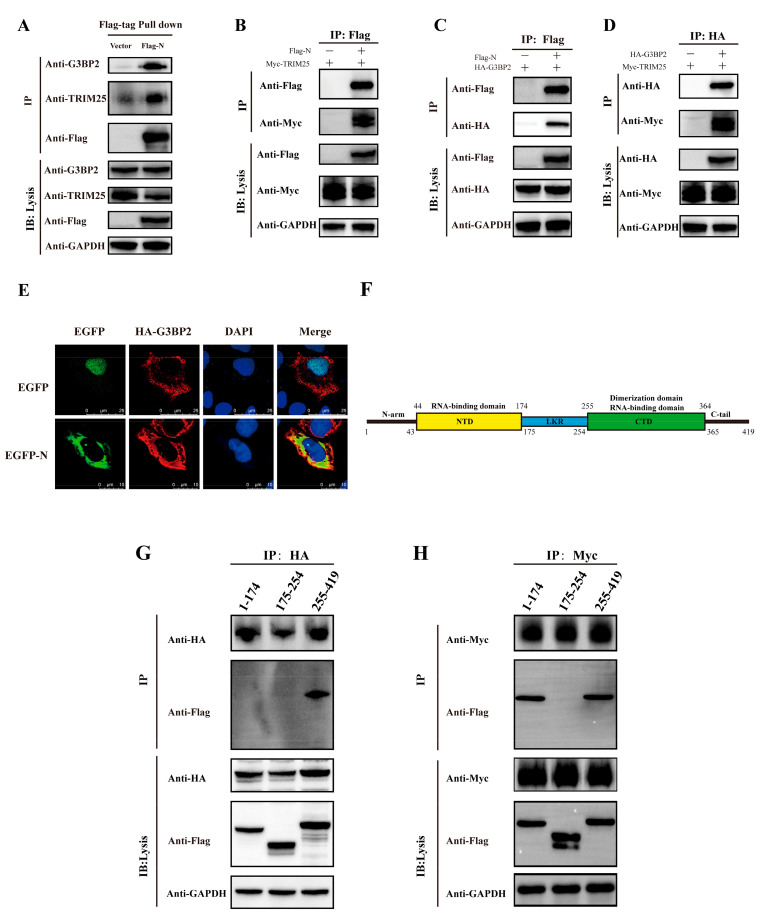
Both the TRIM25 and G3BP2 protein are coimmunoprecipitated by the N protein. (**A**) The pcDNA3.1 vector and pcDNA3.1-N-Flag were transfected into cells. Twenty-four hours after transfection, TRIM25 and G3BP2 were precipitated by the N protein in pulldown experiments using the Flag tag. (**B**) The pcDNA3.1-TRIM25-Myc and pcDNA3.1-N-Flag were cotransfected into cells. Twenty-four hours after transfection, the interaction between the N protein and TRIM25 was detected by a coimmunoprecipitation experiment. (**C**) The pcDNA3.1-G3BP2-HA and pcDNA3.1-N-Flag were cotransfected into cells. Twenty-four hours after transfection, the interaction between N protein and G3BP2 was detected by coimmunoprecipitation. (**D**) pcDNA3.1-G3BP2-HA and pcDNA3.1-TRIM25-Myc were cotransfected into cells. Twenty-four hours after transfection, interaction between TRIM25 and G3BP2 was detected by coimmunoprecipitation. (**E**) Twenty-four hours after pcDNA3.1-G3BP2-HA and pcDNA3.1-N-Flag were cotransfected into cells, a cellular immunofluorescence assay was performed to assess the interaction between the N protein and G3BP2. Red fluorescence corresponds to the G3BP2-positive signal, green fluorescence labels SARS-CoV-2 N, and blue fluorescence corresponds to nuclei (DAPI staining; scale bar, 10 μm). (**F**) Diagram of the N protein divided into three parts: amino acids 1–174, 175–255 and 255–419. (**G**) pcDNA3.1-G3BP2-HA and pcDNA3.1-N (1–174/175–225/255–419)-Flag were cotransfected into cells. Twenty-four hours after transfection, an interaction between the 1–174 or 175–255 or 255–419 region of the N protein and G3BP2 was detected by coimmunoprecipitation. (**H**) pcDNA3.1-TRIM25-Myc and pcDNA3.1-N (1–174/175–225/255–419)-Flag were cotransfected into cells. Twenty-four hours after transfection, an interaction between the 1–174, 175–255 or 255–419 region of the N protein and G3BP2 was detected by coimmunoprecipitation.

**Figure 4 vaccines-10-02042-f004:**
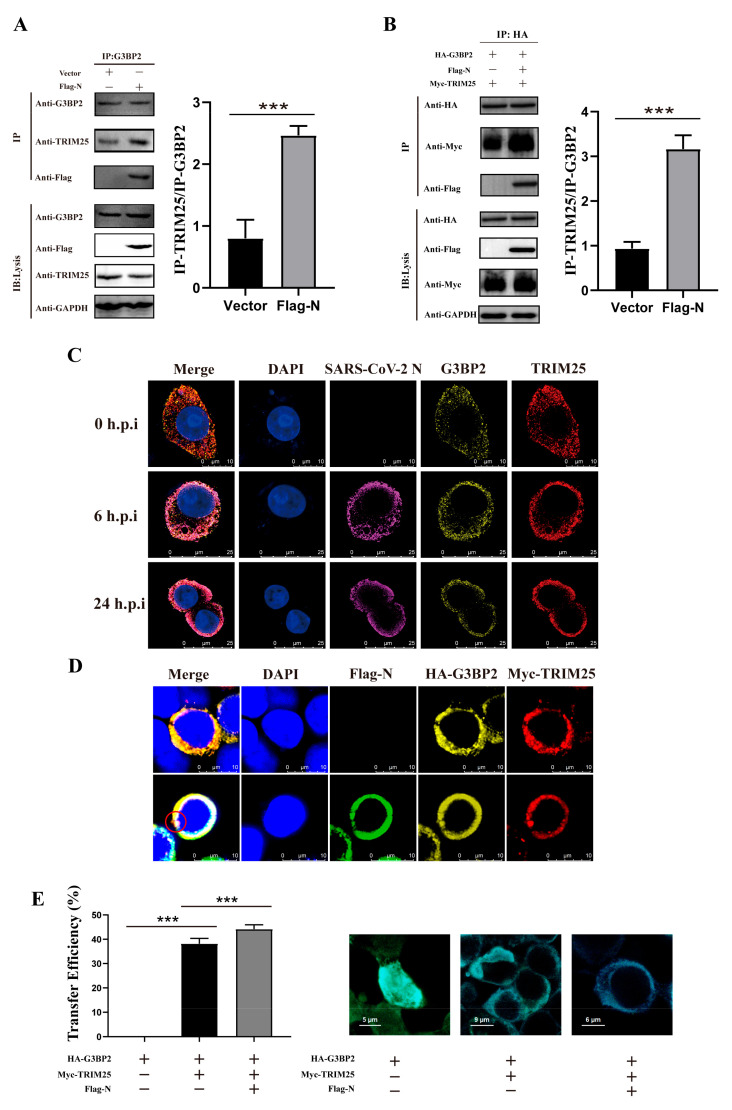
The engagement of G3BP2-TRIM25 binding is enhanced by the recruitment of the N protein. (**A**) The pcDNA3.1 vector and pcDNA3.1-N-Flag were transfected into cells. Twenty-four hours after transfection, coimmunoprecipitation showed that binding of TRIM25 to G3BP2 increased significantly in the presence of N. Grayscale results were analyzed using a Gel-Pro analyzer. Data are presented as means ± SD (paired *t* test, *n* = three biological replicates per group, * 0.01 < *p* < 0.05, ** 0.001 < *p* < 0.01, *** *p* < 0.001). (**B**) pcDNA3.1-TRIM25-Myc, pcDNA3.1-G3BP2-HA and pcDNA3.1-N-Flag were cotransfected into cells. Twenty-four hours after transfection, coimmunoprecipitation showed that binding of TRIM25 to G3BP2 increased significantly in the presence of N. Grayscale results were analyzed using a Gel-Pro analyzer. Data are presented as means ± SD (paired *t* test, *n* = three biological replicates per group, * 0.01 < *p* < 0.05, ** 0.001 < *p* < 0.01, *** *p* < 0.001). (**C**) From 0 to 24 h after cells were infected with SARS-CoV-2, confocal imaging showed that TRIM25 and G3BP2 gradually colocalized with N in the cytoplasm from 6 to 24 h.p.i. Red fluorescence corresponds to the TRIM25-positive signal, yellow fluorescence indicates G3BP2, purple fluorescence corresponds to SARS-CoV-2 N, and blue fluorescence corresponds to nuclei (DAPI staining; scale bar, 10 μm). (**D**) pcDNA3.1-TRIM25-Myc, pcDNA3.1-G3BP2-HA and pcDNA3.1-N-Flag were cotransfected into cells, and confocal imaging showed that TRIM25 and G3BP2 colocalized with N in the cytoplasm. Red fluorescence corresponds to the TRIM25-positive signal, yellow fluorescence labels G3BP2, green fluorescence corresponds to SARS-CoV-2 N, and blue fluorescence corresponds to nuclei (DAPI staining; scale bar, 10 μm). (**E**) We divided the plasmid into three groups, the first group was pcDNA3.1-G3BP2-HA; the second group was pcDNA3.1-TRIM25-Myc and pcDNA3.1-G3BP2-HA; the third group was pcDNA3.1-TRIM25-Myc, pcDNA3.1-G3BP2-HA and pcDNA3.1-N-Flag was transfected into 293T cells. Twenty-four hours later, the cell samples were collected for immunofluorescence staining. Samples were analyzed by fluorescence resonance energy transfer analysis software. Data are presented as means ± SD (paired *t* test, *n* = 5 biological replicates per group, *** *p* < 0.001).

**Figure 5 vaccines-10-02042-f005:**
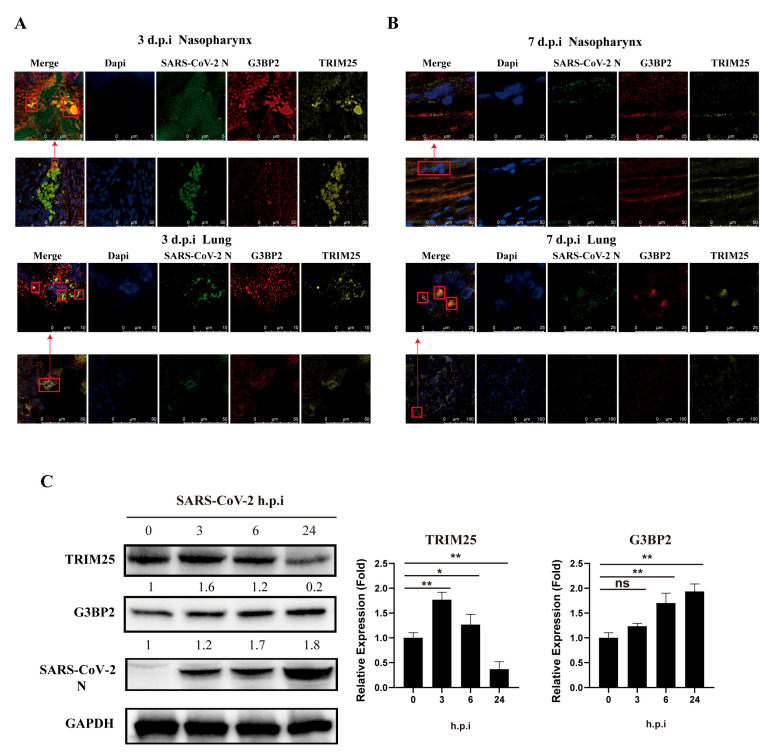
Colocalization of G3BP2, TRIM25 and N in the nasopharynx and lung tissue of SARS-CoV-2-infected rhesus monkeys. (**A**) Tissue sections of nasopharynx and lung samples of rhesus macaques were collected at 3 dpi and stained with a fluorescence-labeled antibody for confocal observation. Yellow fluorescence corresponds to the TRIM25-positive signal, red fluorescence labels correspond to G3BP2, green fluorescence corresponds to SARS-CoV-2 N, and blue fluorescence corresponds to nuclei (DAPI staining; scale bar, 10 μm or 50 μm). (**B**) Tissue sections of the nasopharynx and lung samples of rhesus macaques were collected at 7 dpi and stained with a fluorescence-labeled antibody for confocal observation. Yellow fluorescence corresponds to the TRIM25-positive signal, red fluorescence labels correspond to G3BP2, green fluorescence corresponds to SARS-CoV-2 N protein, and blue fluorescence corresponds to nuclei (DAPI staining; scale bar, 10 μm or 50 μm). (**C**) 16HBE cells (MOI = 0.5) were infected with SARS-CoV-2. Cell samples were collected 0, 3, 6, and 24 h after infection for western blot assays. Expression of G3BP2 and TRIM25 increased gradually in the early stage of infection. (paired *t* test, *n* = 3 biological replicates per group, ns > 0.05, * 0.01 < *p* < 0.05, ** 0.001 < *p* < 0.01).

**Figure 6 vaccines-10-02042-f006:**
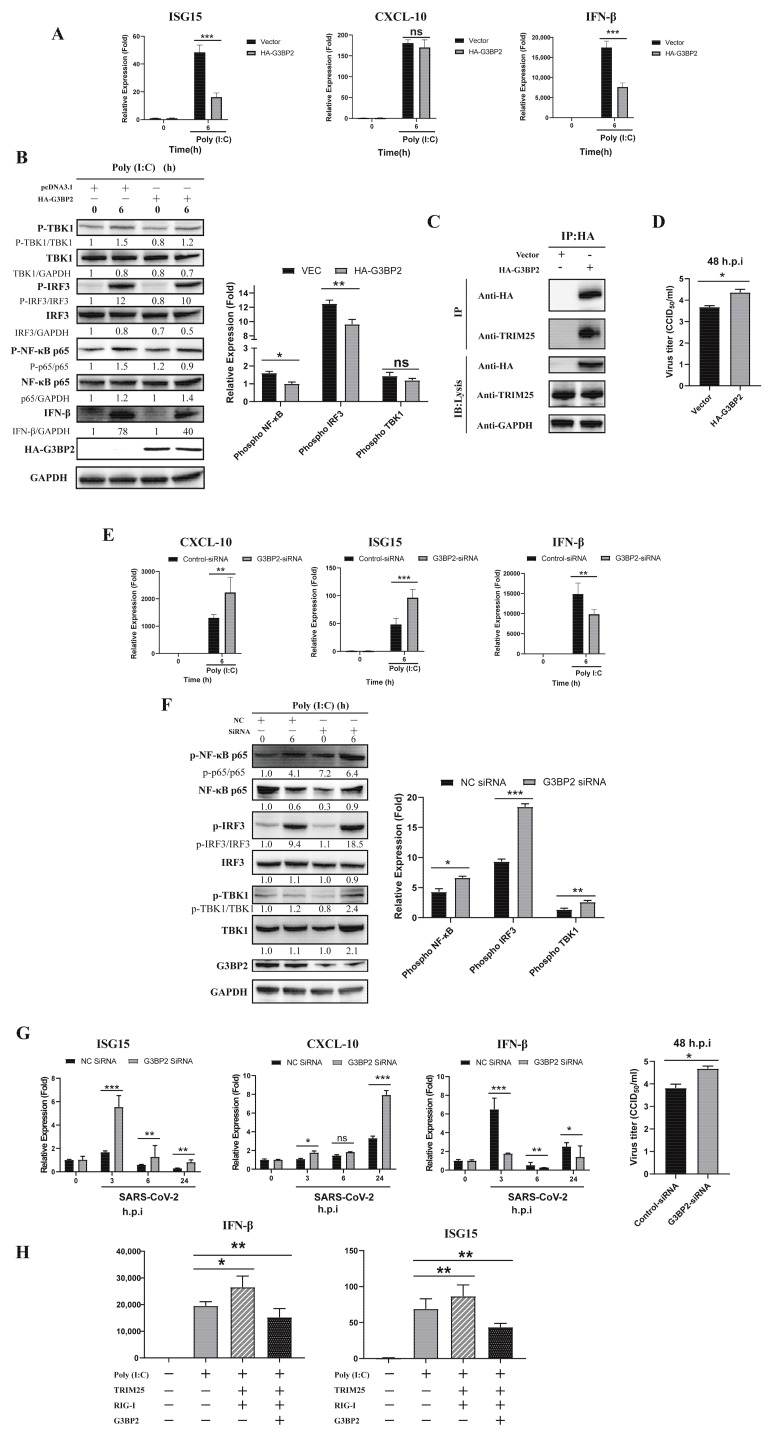
Overexpression of G3BP2 impairs TRIM25 regulation of the type I interferon signaling pathway during SARS-CoV-2 infection. (**A**) The pcDNA3.1 vector and pcDNA3.1-G3BP2-HA were transfected into A549 cells. Twenty-four hours after transfection, gene expression of ISG15, CXCL10 and IFN-β was detected by real-time PCR after stimulation with poly (I:C). Data representing the relative expression of different genes are shown as means ± SD (paired *t* test, *n* = 3, biological replicates per group, ns > 0.05, * 0.01 < *p* < 0.05, ** 0.001 < *p* < 0.01, *** *p* < 0.001). (**B**) The pcDNA3.1 vector and pcDNA3.1-G3BP2-HA were transfected into A549 cells. Twenty-four hours after transfection, phosphorylation was detected by western blot analysis of the IRF3, NF-κB and TBK1 proteins from 0- and 6-h cell samples after stimulation with poly (I:C). Grayscale results were analyzed using a Gel-Pro analyzer. Data are presented as means ± SD (paired *t* test, *n* = three biological replicates per group, ns > 0.05, * 0.01 < *p* < 0.05, ** 0.001 < *p* < 0.01, *** *p* < 0.001). (**C**) The pcDNA3.1 vector and pcDNA3.1- G3BP2-HA were transfected into A549 cells. Twenty-four hours after transfection, coimmunoprecipitation showed that overexpressed G3BP2 bound to intracellular TRIM25. (**D**) The pcDNA3.1 vector and pcDNA3.1- G3BP2-HA were transfected into A549 cells. Twenty-four hours after transfection, SARS-CoV-2 was inoculated into the cells; the virus titer in cells overexpressing G3BP2 was higher than that of control cells at 48 h.p.i. (**E**) G3BP2 siRNA and control siRNA were transfected into 16HBE cells. After 48 h, cell samples were collected at 0 and 6 h after poly (I:C) stimulation, and the expression of ISG15, CXCL10 and IFN-β was detected by real-time PCR. Data representing relative expression that normalized by β-actin of different genes are shown as means ± SD (paired *t* test, *n* = 3, biological replicates per group, ns > 0.05, * 0.01 < *p* < 0.05, ** 0.001 < *p* < 0.01, *** *p* < 0.001). (**F**) G3BP2 siRNA and control siRNA were transfected into 16HBE cells. After 48 h of poly (I:C) stimulation, phosphorylation of IRF3, NF-κB and TBK1 in 0- and 6-h cell samples was detected by western blotting. Grayscale results were analyzed using a Gel-Pro analyzer. Data are presented as means ± SD (paired *t* test, *n* = three biological replicates per group, * 0.01 < *p* < 0.05, ** 0.001 < *p* < 0.01, *** *p* < 0.001). (**G**) G3BP2 siRNA and control siRNA were transfected into 16HBE cells. At 48 h after SARS-CoV-2 infection, cell samples were collected at 0 and 6, and expression of ISG15, CXCL10 and IFN-β was detected by real-time PCR. Data represent the relative expression of different genes. The virus titer in G3BP2-knockdown cells was higher than that of control cells at 48 h.p.i. Data are presented as means ± SD (paired *t* test, *n* = 3, biological replicates per group, * 0.01 < *p* < 0.05, ** 0.001 < *p* < 0.01, *** *p* < 0.001). (**H**) HA-G3BP2, Myc-TRIM25 and Flag-RIG-I were transfected into A549 cells. Twenty-four hours after transfection, the cells were stimulated with poly (I:C) and harvested after 6 h. The expression of IFN-β and ISG15 was determined by real-time PCR. Data are presented as means ± SD (paired *t* test, *n* = three biological replicates per group, * 0.01 < *p* < 0.05, ** 0.001 < *p* < 0.01).

**Figure 7 vaccines-10-02042-f007:**
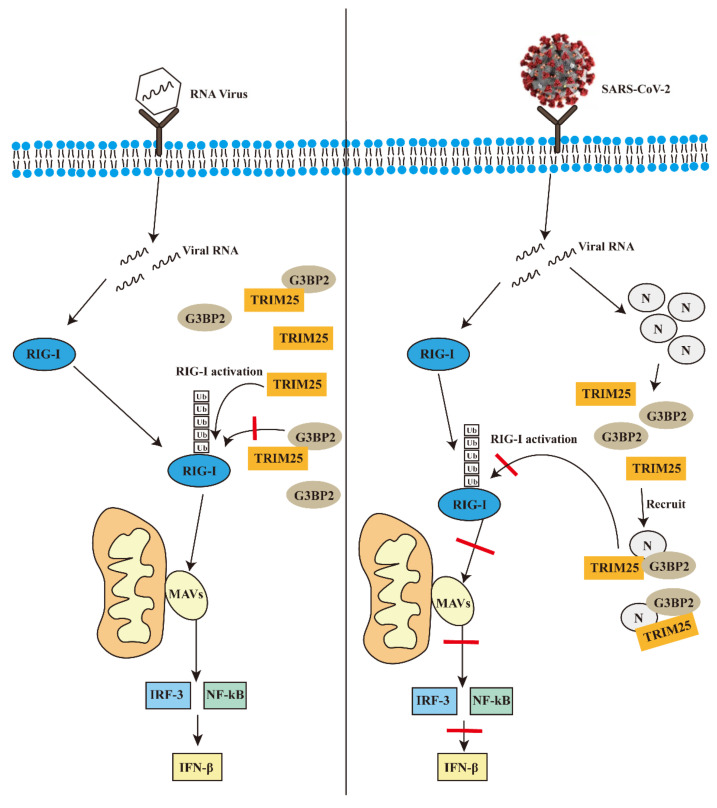
The possible mechanism of the N protein of SARS-CoV-2 promotes the interaction between G3BP2 and TRIM25 to inhibit innate immunity during infection.

## Data Availability

Data sharing not applicable to this article as no datasets were generated or analyzed during the current study.

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
