# Peer review of "Engagement of the G3BP2-TRIM25 Interaction by Nucleocapsid Protein Suppresses the Type I Interferon Response in SARS-CoV-2-Infected Cells"

_vaccines, 2022, doi:10.3390/vaccines10122042_

Round 1

Reviewer 1 Report

Yang et al report on the contribution of the SARS-CoV-2 N protein to suppression of the IFN response.  They investigate the impact of SARS-CoV-2 infection on RIG-I-dependent signal transduction and the role of N therein, corroborating previous reports. In subsequent experiments, the authors verify a direct interaction of N with TRIM25 and newly describe an interaction of N with G3BP2. Further analysis revealed that N is able to enhance the TRIM25-G3BP2 interaction. All three proteins are shown to colocalize in cells as well as in infected monkeys. Finally, the authors investigate the impact of the TRIM25-G3BP3 interaction on RIG-I signaling by overexpression and knockdown experiments. Taken together, the data are compatible with the model that N recruits G3BP2 to enhance a negative feedback inhibition of TRIM25 and thereby weakens RIG-I signalling.

In summary, the paper presents important data arguing for a novel mechanism by which N contributes to SARS-CoV-2 inhibition of RIG-I signalling. The paper is clearly written and merits publication.

Specific comments:

1. p2, line71: the SARS-CoV-2 virus strain should be provided in more detail (wt, delta, omicron strain?)

2. p4, line 166: the proteins investigated in this figure are downstream of RIG-I

3. p7, line 228: the text does not match the figure legend

4. p8, line 270: the experiments described in Fig. S1 would argue for an effect of N on the CMV-promoter driven in the pcDNA plasmid. However, they do not suggest an N-promoted expression of TRIM25.

5. Figure 6, E and G: what means “relative expression”? Did the authors use a housekeeping gene for normalization? The authors should  describe this in more detail in the legend.

6. p.14, line 393: when a distinct result of a previous study is discussed the corresponding study should be cited at this passage in the text.

Author Response

  1. p2, line71: the SARS-CoV-2 virus strain should be provided in more detail (wt, delta, omicron strain?)

A:Thank you very much for your suggestion. We have added the virus strain information in the revised draft.

  1. p4, line 166: the proteins investigated in this figure are downstream of RIG-I

A: Thank you very much for your suggestion. We have revised the corresponding text.

  1. p7, line 228: the text does not match the figure legend

A: Thank you very much for your question. I'm sorry that our description has misunderstood you. We have modified the text.

  1. p8, line 270: the experiments described in Fig. S1 would argue for an effect of N on the CMV-promoter driven in the pcDNA plasmid. However, they do not suggest an N-promoted expression of TRIM25.

A: Thank you very much for your question. In fact, this is just an interesting phenomenon we found, so we just briefly described this phenomenon with supplement figure.

  1. Figure 6, E and G: what means “relative expression”? Did the authors use a housekeeping gene for normalization? The authors should  describe this in more detail in the legend.

A:Thank you very much for your suggestion, which we have added in the legend.

  1. 14, line 393: when a distinct result of a previous study is discussed the corresponding study should be cited at this passage in the text.

A: thank you very much for your suggestion. In fact, the sentence in which the citation is located is a description of our previous study, and we have adjusted the location of the citation.

Reviewer 2 Report

In the manuscript entitled “Engagement of the G3BP2-TRIM25 Interaction by Nucleocapsid Protein Suppresses the Type I Interferon Response in SARS-CoV-2-Infected Cells” the authors showed that G3BP2 is involved in the process of N protein binding to TRIM25, which weakens the innate immune response. This is an interesting and well-written manuscript. The data are clearly presented, the material and methods are fully described and the conclusions are compelling. There are, however, some minor suggestions that need to be incorporated before the manuscript will be accepted for publication;

Comments:

1)     Authors mentioned J-2 staining for double-stranded RNA in Fig 1C, but didn’t mention any such details within the material and method section. Although the magnifications are the same for all the selected images (scale bar-25 µm), it seems that the nuclei size is decreasing gradually with time (as the scale bar is not so clear). It would be nice to select the same size scale bar for all the images shown in different panels of the same figure (wherever applicable). Also, at 6 hours and 24 hours post-infection with SARS-CoV-2, the signals from SARS-CoV-2-S RNA are very faint (green fluorescence labels) as compared to SARS-CoV-2 N (yellow) signal. It would be great to add an additional negative control like heat-inactivated SARS-CoV-2 to show if there are any non-specific signals. It would be great if author’s can address all these points.

2)     It would be great to add a schematic representation showing the possible mechanism involved and summarize the findings of this manuscript.

Minor Comments:

3)     Line 19: interacting ---à interaction

4)     Line-71; Material and Method section; Please mention the full details of the SARS-CoV-2 strains used in this study.

5)     Line-75; Material and Method section; final concentration for one multiplicity of infection (MOI=0.5)---à The statement is confusing, please correct this line.

Author Response

  • Authors mentioned J-2 staining for double-stranded RNA in Fig 1C, but didn’t mention any such details within the material and method section. Although the magnifications are the same for all the selected images (scale bar-25 µm), it seems that the nuclei size is decreasing gradually with time (as the scale bar is not so clear). It would be nice to select the same size scale bar for all the images shown in different panels of the same figure (wherever applicable). Also, at 6 hours and 24 hours post-infection with SARS-CoV-2, the signals from SARS-CoV-2-S RNA are very faint (green fluorescence labels) as compared to SARS-CoV-2 N (yellow) signal. It would be great to add an additional negative control like heat-inactivated SARS-CoV-2 to show if there are any non-specific signals. It would be great if author’s can address all these points.

A: thank you very much for your question. In fact, Jmur2 dye is also an antibody, so its staining can follow the steps of the immunofluorescence experiment. The problem of observing the infection process is not to observe the infection process of the same cell, but to collect the cell samples at different time points after infection for immunofluorescence staining, so the size of the nucleus in the picture will be different. The problem of fluorescence intensity may be related to antibodies or fluorescent dyes, but it does not affect the judgment of the experimental results. With regard to the negative control, we regret that there are not enough experimental conditions to supplement the experiment, but we believe that the current results can support our conclusion.

It would be great to add a schematic representation showing the possible mechanism involved and summarize the findings of this manuscript.

A: Thank you very much for your suggestion, and we have added the corresponding results to the conclusion section.

Minor Comments:

  • Line 19: interacting ---à interaction

A: thank you very much for pointing out our mistake. We have corrected it.

  • Line-71; Material and Method section; Please mention the full details of the SARS-CoV-2 strains used in this study.

A: Thank you very much for your suggestion. We have added the relevant information.

  • Line-75; Material and Method section; final concentration for one multiplicity of infection (MOI=0.5)---à The statement is confusing, please correct this line.

A: Thank you very much for your suggestion. We have revised the relevant text.